# Effect of Barley Variety on Feed Intake, Digestibility, Body Weight Gain and Carcass Characteristics in Fattening Lambs

**DOI:** 10.3390/ani11061773

**Published:** 2021-06-14

**Authors:** Mulugeta Tilahun Keno, Taye Tolemariam, Solomon Demeke, Jane Wamatu, Ashraf Alkhtib, Geert P. J. Janssens

**Affiliations:** 1College of Agriculture and Veterinary Medicine, Jimma University, P.O. Box 378 Jimma, Ethiopia; taye.tolemariam@ju.edu.et (T.T.); solomondemeke2000@gmail.com (S.D.); 2Department of Nutrition, Genetics, and Ethology, Faculty of Veterinary Medicine, Ghent University, Heidestraat 19, 9820 Merelbeke, Belgium; Geert.Janssens@ugent.be; 3International Centre for Agricultural Research in Dry Areas, P.O. Box 5689 Addis Ababa, Ethiopia; J.Wamatu@cgiar.org; 4School of Animal, Rural and Environmental Sciences, Brackenhurst Campus, Nottingham Trent University, Southwell NG25 0QF, Nottinghamshire, UK; ashraf.alkhtib@ntu.ac.uk

**Keywords:** crop residue, variety, fattening performance, food, feed

## Abstract

**Simple Summary:**

Using a native barley straw as reference, the barley straw from four improved varieties were tested on digestibility and performance in lambs. Differences were observed on feed intake, digestibility, body weight gain and feed-to-gain ratio among lambs fed straws from different barley varieties, pointing to the importance of genetic variation in the feeding value of barley straw.

**Abstract:**

Twenty lambs (18 ± 0.22 kg initial weight) were blocked by weight and individually assigned into pens to evaluate the effects of barley straw variety on digestibility, growth performance and carcass characteristics. The following four treatments were tested: (1) a local barley straw (as control), (2) HB1963 (high grain and straw yields), (3) Traveller (high straw yielder), and (4) IBON174/03 (high grain yielder). A concentrate (50:50 wheat bran and noug seed cake) was offered constantly (300 DM g), whereas the straw was offered ad libitum. The digestibility trial lasted 22 days (15 days to adapt to dietary treatments and 7 days for sampling). The growth performance trial lasted 90 days. At the end, all of the lambs were slaughtered, and their carcasses were evaluated. The IBON174/03 variety had a higher (*p* < 0.05) intake of organic matter and crude protein, a higher dry matter and organic matter digestibility than the control, and a faster growth than the control. The feed-to-gain ratio was similar among treatments. The slaughter and empty body weights of lambs in the IBON174/03 group were higher than the control variety (*p* < 0.05). The present study showed that the feeding value of barley straw can differ substantially between varieties and therefore must be considered in the choice of a barley variety.

## 1. Introduction

Barley (*Hordeum vulgare L.*) is a multiple-purpose crop with high economic and social importance. It is grown to produce grain for human and livestock consumption and malt for brewing [1,2]. The breeding and selection of barley has been focusing on the optimization of grain production, without due consideration of the yield and quality of straw as livestock feed. Newly improved varieties and cultivation methods have led to a decrease in straw yields [3,4,5]. Improved varieties have been rejected because of poor straw traits in crops including barley [6,7] and finger millet [8]. In mixed crop-livestock farming systems, the use of crop residues for livestock feeding is important due to the expansion of cropland and the subsequent productivity decline of natural pastures [9]. The authors of [10] reported that straw has become an important part of total crop value. The contribution of genetic as opposed to non-genetic factors to grain and fodder yields and to straw digestibility varies between crop species and between the varieties within a crop species. Varietal differences in the chemical composition and feeding value of crop residues have been reported in wheat, rice, sorghum and maize [11]. The authors of [12] showed the high genotypic variability in grain yield, straw yield and the nutrient composition of straw in naked barley landraces. The authors of [6,13,14,15] found a varietal variation in the intake and nutrient digestibility of barley straw when it was fed to sheep. A significant effect of genotype, row type and morphology was observed on the nutritive value of barley straw [13,14]. The chemical composition and ruminal fermentability of barley straw was significantly affected by the planting date, the irrigation level and the variety [15]. The effects of variety on the performance of straw-fed animals still need to be determined.

The objective of this study was to evaluate the effect of straw from different barley varieties on the feed intake, digestibility, body weight gain and carcass characteristics of Horro lambs. We hypothesized that the digestibility and growth performance in straw-fed sheep can add crucial information to the decision-making process when selecting the optimal barley variety for dual-purpose use.

## 2. Material and Methods

### 2.1. Animal Care

Animal care, handling and maintenance throughout the experiment were in accordance with the animal welfare regulations of Jimma University.

### 2.2. Study Sites and Plant Materials

The feeding trial experiment was conducted at Jimma University, College of Agriculture and Veterinary Medicine (longitude, 7°40′ N 36°50′ E; latitude, 7.667° N 36.833° E; elevation, 1780 m; average temperature 29 °C). Barley varieties were grown at Kolumsa Agricultural Research Center, Kofele site, located in the West Arsi zone of Oromia Regional State, Ethiopia. The center is located 280 km, southwest of Addis Ababa, the capital city of Ethiopia at 06°50′ to 07°09′ N latitude and 38°38′ to 39°04′ E longitudes and at an altitude of 2650 m above sea level. The average annual maximum and minimum temperatures are 21 and 4 °C, respectively. The average annual rainfall is 950.6 mm. Soil type is loamy and acidic [10]. Three improved varieties were initially selected from 40 food and malt barley varieties that had been evaluated at Kofele and Bekoji, Ethiopia, under the National Variety Trials of the Ethiopian Barley Improvement Program. The varieties, IBON174/03, Traveller and HB1963, were selected as a high grain yielder, a high straw yielder and food-feed (high in grain yield as well as straw yield), respectively. The three selected improved varieties and one local (control) were then planted at the Kofele site in Ethiopia. All varieties received the same agronomic practices as per recommendations for barley growing in Ethiopia. The above-ground biomass of each plot was manually harvested at physiological maturity, air-dried for two weeks to a constant moisture, then threshed. The straw was chopped to a theoretical length of 2 cm, put in plastic bags and stored for one month until the start of the feeding trial.

### 2.3. Animals, Experimental Design and Diets

Twenty Horro yearling lambs with an initial body weight of 18.0 ± 0.2 kg were obtained from a local market. The Horro breed is mainly maintained for meat in the study area. The lambs were quarantined for three weeks. Experimental lambs were vaccinated against ovine pasteurellosis using a *Pasteurella maltocida* type A vaccine and sheep pox using a live lyophilized capripox vaccine and dewormed against external and internal parasites using ivermectin. Based on their initial weight, the lambs were arranged into four groups, each with five lambs, in a randomized complete block design. They were placed in individual pens; the pens were 2 m long and 1.5 m wide with concrete floors, an open-air platform and equipped with a drinking and feeding trough in a randomized complete block design. The following four treatments were tested: (1) a local straw barley (as control), (2) HB1963 (high grain and straw yield), (3) Traveller (a high straw yielder), and (4) IBON174/03 (a high grain yielder). A concentrate (50:50 wheat bran and noug seed cake) was offered at a fixed amount (300 gDM/d), whereas the straw was offered ad libitum. Description of the selected varieties are presented in Table 1.

Lambs were fed twice a day (0800 h and 1600 h) in equal proportions. Lambs had free access to a salt lick and clean drinking water.

### 2.4. Digestibility Trial

The growth performance trial was conducted 10 days after the digestibility experiment. There were 15 days for adaptation to the experimental conditions and feeds, followed by the total collection of feces for seven consecutive days. The daily feed offered and refused per lamb was collected. Total fecal output was collected by daily emptying of every fecal collection bag in the morning, prior to offering feed and water. Feces were weighed fresh, thoroughly mixed and 20% of the feces were sampled per lamb and stored in a freezer at −18 °C. Samples were pooled per lamb over the collection period and 20% of the composite sample was taken, weighed, and partially dried at 60 °C for 72 h. The apparent digestibility of dry matter (DM) and other nutrients were determined as a percentage of the nutrient intake not recovered in the feces.

### 2.5. Growth

The 90-day feeding and growth experiment was conducted after the completion of the digestibility trial and after a 10-day rest period. The lambs were fed the same treatment during the digestibility and growth trials. The live weight of the lambs was recorded at the start of the trial, and every 10 days subsequently after overnight fasting and before morning feeding, using a hanging scale with a sensitivity of 100 g, for 90 consecutive days and a KERN Scale EWJ 6000 g with a sensitivity of 1 g was used to weigh the feed and refusals. The daily feed offered and the refusals were weighed and recorded per sheep. Daily feed and nutrient intakes were calculated as the difference between the offered feed and the refusals on a DM basis. Average daily gain (ADG) was calculated as the difference between the final and initial weights divided by the number of feeding days. The feed-to-gain ratio (FGR) was calculated as the total DMI to the ADG. Samples of the feed offered were collected per batch, whereas samples of the refusals were taken daily from each lamb and stored in plastic bags. Subsamples of offered feed and refusals were dried at 60 °C for 48 h, then ground to pass through a 1-millimeter screen and stored for chemical analysis.

### 2.6. Carcass Evaluation

At the end of the experiment, all lambs were slaughtered after 24 h of fasting to determine the treatment’s effects on carcass’ characteristics. Lambs were individually weighed before slaughter. Carcass variables were registered individually. Slaughtering was performed as described by [16]. The weights of the head with tongue, feet, skin, blood, liver and gall bladder, heart, kidneys with fat, lungs and trachea, abdominal fat, testicles and other genitalia, and full and empty gastrointestinal tracts were recorded. Empty body weight (EBW) was calculated as the slaughtered body weight minus gastro-intestinal tract’s contents. Hot carcass weight (HCW) was determined as the body after removing the skin, head, forefeet, hind feet and all the viscera and fat deposits. Dressed carcasses were weighed within 1 h and recorded as hot carcass weight and then chilled for 24 h at 4 °C, weighed again and recorded as cold carcass weight. The dressing percentage on a slaughter body weight basis and an empty body weight basis was calculated as the percentage of hot carcass weight to slaughter body weight and empty body weight.

### 2.7. Chemical Analysis

All feed and feces samples were analyzed for dry matter (DM) [17] (method 934.01); [17] ash (method 942.05); nitrogen [17] (method 954.01); neutral detergent fiber (NDF), which was analyzed using the procedure [18]; and acid detergent fiber (ADF) [18]. Crude protein content was calculated as N × 6.25.

### 2.8. Statistical Analysis

The experimental lambs were blocked according to live weight. Data of the current study were analyzed according to the following model:Y_ij_ = μ + T_i_ + Bj + E_ij_(1)
where Y_ij_ is the response variable, μ is the overall mean, T_i_ is the effect of treatment, B_i_ is the effect of block and E_ij_ is the residual. Treatment means were separated using the Tukey test at *p* < 0.05. The statistical analysis was performed using SPSS [14,19].

## 3. Results

### 3.1. Chemical Composition of the Experimental Diet

The tested barley varieties contained relatively more CP than the local straw (control). Numerically, the HB1963 variety was higher in NDF, ADF, ADL and ash concentrations than the other varieties (Table 2).

### 3.2. Nutrients Intake and Digestibility

The sheep ate more dry matter and protein from the high grain yielder (IBON174/03) than from the food-feed variety (HB1963) (Table 3). The organic matter intake was higher from the IBON174/03 than from the HB1963. The lambs on the IBON174/03 treatment consumed 353 g/d organic matter from straw and 277 g/d organic matter from concentrates. The lambs on the HB1963 treatment consumed 289 g/d organic matter from straw and 277 g/d organic matter from concentrate. The dry matter and organic matter digestibility of straw were higher with the IBON174/03 than with Traveller and the local variety (control), whereas the greater crude protein digestibility observed with the control treatment compared to the other varieties was a reflection of a lower straw intake (325 g/d) and the lower protein content (4.3%) in the local straw. For example, in the local straw treatment, lambs consumed 14 g of protein from straw and 61 g of protein from concentrates (19% of the total protein consumed was low-digestible protein), whereas the lambs fed IBON174/03 consumed 20.5 g of protein from straw and 61 g of protein from concentrates (25% of the total protein consumed was low-digestible protein) (Table 3). No difference was observed in the NDF and ADF intake and the digestibility between the varieties.

### 3.3. Growth Performance and Carcass Characteristics

The average daily gain was higher for lambs on the IBON174/03 treatment compared to the control. A higher intake was observed for this group than the HB1963 group. The feed-to-gain ratio did not differ between the varieties but IBON174/03 led to faster growth than HB1963 and the local variety, resulting in a higher slaughter and empty body weight than the other varieties (Table 4). None of the carcass components differed between the varieties (Table 4).

## 4. Discussion

The crude protein (CP) concentration of the studied barley varieties ranged from 4.3% in the local variety to 5.5% in IBON174/03, which resembles the values of [20]. The observed value was below the range of 7–7.5% assumed to be sufficient for the maintenance and rumen microbial function of ruminant animals [21]; therefore, supplementation with concentrate feed with a high protein content is important to fulfill the protein requirement of animals. All of the tested varieties had a high fiber content (higher in HB1963, lower in IBON174/03), similar with the result of [22].

Based on the reported DM digestibility (DMd) for wheat bran (76%) and noug seed cake (86%) [20,23,24], the DMd for the concentrates used in this trial was 80.5%. When combining this figure with the proportion in the actual diets, the DM digestibility of straw can be estimated as follows:

(DM intake × digestibility DM) − (0.805 × 300)/straw intake. Straw intake was 325, 312, 356, and 372 g for local straw, HB1963, Traveller and IBON174/03, respectively. This calculation renders a 48% DMd for local straw, which is in close agreement with the digestibility of barley straw determined in a previous report [25]. The estimated DMd for HB1963, Traveller and IBON174/03 are 56.5, 52.3 and 64.9%, respectively, confirming a higher digestibility of the selected straw varieties compared with the non-selected local variety. Although straw digestibility was not a selection target, this feature has been improved through selection.

The higher digestibility coefficient of the total diet in this study was thus due to the combination of straw with a protein-rich concentrate feed (300 g DM/day/lamb). Dietary protein enhances microbial proliferation in the rumen, which enables rumen fermentation [26]. The higher apparent DM and OM digestibility of the rams fed IBONE174/03/ straw was probably due to the high leaf-to-stem ratio (Table 1) and its lower NDF and ADL content compared to other barley varieties, since it is mainly fiber that influences digestibility [27]. The DM and OM digestibility of the lambs fed Traveller straw were probably lower due to its higher NDF and ADF content.

The fiber itself was not better digested (at least not significantly), but it is likely that the lower fiber content improved the accessibility of rumen microbiota and digestive enzymes to their substrates. This hypothesis is supported by the lower protein digestibility in the rams fed Traveller straw that also had the highest fiber concentrations. The negative correlation between the fiber concentration of the straw and the DM intake indicates that fiber concentration in the diet was reducing the voluntary feed intake.

The high voluntary DM intake of the lambs fed IBON174/03 straw might be due to lower fiber content and high leaf-to-stem ratio compared to the other straws. The authors of [28,29] demonstrated that high fiber induced a low digestibility and voluntary feed intake, which is in line with the current study. The greater overall feed intake in lambs fed IBON174/03 straw, did not imply a large intake of fiber because the difference in the fiber concentration was compensated by the difference in intake. Despite the higher dietary fiber content in the current study, the DM feed intake per kg of body weight in the current study was in the recommended range of dry matter intake for ruminants (2–6% of body weight) [22].

The observed differences in the average daily gain between the treatments might originate from differences in the intake and nutrient digestibility. The higher intake in the high grain yielder group was demonstrated, but also the higher DM and OM digestibility will have added to the higher growth performance in this group. Since the digestibility of fiber and protein were not higher, and fat content is very low in straw, we postulate that the leafier material in the high grain yielder straw allowed a faster ruminal escape of the starch in the concentrates, leading to the more efficient enzymatic digestion compared to fermentation. It has been demonstrated that leafy material has a faster ruminal escape than stem material in sheep [30].

The numerically higher feed-to-gain ratio for the high grain yielder straw (IBON174/03) agrees with this improved efficiency. This hypothesis must be confirmed through measuring ruminal passage, which we were unfortunately unable to perform. It may signify that the effect of the barley variety on the utilization of a straw-based diet depends on the composition of the total ration, an aspect that warrants further study.

The higher slaughter weight on the IBON174/03 straw diet is an evident outcome of an increased intake and digestibility. The greater carcass yield with the rams fed IBON174/03 was mainly a direct effect of greater growth, since the dressing percentage was only numerically altered. The fact that only a few body parts showed significant differences between the treatments indicates that the better performance with IBON174/03 is a direct effect of the increased intake and digestibility, without apparent changes in the confirmation of the body. The dressing percentage of the sheep observed in the current study was low (33.5 to 35.7%) compared to the report of [31] for the same Horro breed (36.7 to 42.5%). The present study showed that there was no significant difference in the internal organs among the treatments. Internal organs are more affected by the age, breed and sex of the animals, rather than the type of nutrition [32].

The high demand for barley straw resources in the mixed farming systems was already reported by [33,34]. The grain and straw yield of the local barley variety is low (4 t/ha of grain, 4 t/ha of straw) compared to the improved varieties, for example IBON174/03 (7.1 t/ha of grain and 7.5 t/ha of straw), while the population of humans and livestock in the mixed system is increasing.

Generally, this study shows the possibility for choosing barley varieties based on their straw quality in addition to grain yield. This feature enables the use of all of the produced plant biomass to meet the high demand of grain for human consumption as well as straw for livestock feeding in the mixed farming systems of Ethiopia and other tropical countries. The best performing group in this study was fed a IBON174/03 variety.

## 5. Conclusions

In conclusion, the growth performance of sheep can depend on the barley variety that provided the straw in their diet. In particular, the IBON174/03 barley variety was the most promising in terms of the feeding value of the straw, hence it could be recommended as a more suitable candidate in the study area. The including straw quality as a selection criterion for barley can help in enhancing livestock productivity in addition to grain yield for human consumption. This study showed the importance of barley variety when straw is a substantial part of a ruminant’s diet, such as in tropical conditions.

## Figures and Tables

**Table 1 animals-11-01773-t001:** Description of the barley varieties used in the study.

Variety	Grain Yield (t/ha)	Straw Yield (t/ha)	Leaf/Stem (%)
IBON174/03 (High grain yielder)	7.1	7.5	44.7
TRAVELLER (High straw yielder)	6.0	9.1	32.4
HB1963 (Food-feed)	6.4	8.4	38.5
Local (Not improved)	4	4.5	35.3

**Table 2 animals-11-01773-t002:** Nutrient composition of the barley straw varieties and concentrate mixture used in the study.

	IBON174/03	TRAVELLER	HB1963	Local	Concentrate
Dry matter (%)	90.4	91.2	91.3	90.7	91.9
Crude protein (%)	5.5	5.1	5.2	4.3	20.4
Neutral detergent fiber (%)	73.2	77.0	79.3	77.5	47.7
Acid detergent fiber (%)	51.2	55.4	57.7	55.6	23.6
Acid detergent lignin (%)	9.2	11.3	11.7	9.8	8.8
Ash (%)	5.2	7.2	7.4	6.7	7.6

**Table 3 animals-11-01773-t003:** Nutrient intake and nutrient digestibility coefficients in Horro lambs fed diets containing straw from different varieties of barley supplemented with a concentrate mixture.

	IBON174/03	TRAVELLER	HB1963	Local	SEM	*p*
Intake (g/d)						
Straw dry matter	372 ^a^	356 ^ab^	312 ^b^	325 ^ab^	13.8	0.036
Concentrate mix	300	300	300	300	300	
Total dry matter	672 ^a^	656 ^ab^	612 ^b^	625 ^ab^	13.8	0.036
Organic matter	630 ^a^	607 ^ab^	567 ^b^	580 ^ab^	12.9	0.021
Crude protein	81.5 ^a^	79.6 ^ab^	77.4 ^b^	75.3 ^bc^	0.65	0.001
Neutral detergent fiber	416	417	391	395	10.7	0.243
Acid detergent fiber	261	268	251	251	7.7	0.367
Digestibility (%)						
Dry matter	71.9 ^a^	65.2 ^b^	68.3 ^ab^	63.7 ^b^	1.5	0.011
Organic matter	73.4 ^a^	68.7 ^b^	70.8 ^ab^	66.8 ^b^	1.4	0.033
Crude protein	67.5 ^ab^	66.1 ^ab^	59.9 ^b^	68.3 ^a^	1.94	0.040
Neutral detergent fiber	63.8	60.7	67.7	60.5	1.75	0.041
Acid detergent fiber	62.3	58.3	64.4	57.2	1.81	0.051

^a,b,c^ Different superscripts indicate significant differences at *p* < 0.05.

**Table 4 animals-11-01773-t004:** Body weight change and carcass characteristics of Horro lambs fed diets containing straw from different varieties of barley supplemented with a concentrate mixture.

	IBON174/03	TRAVELLER	HB1963	Local	SEM	*p*
Growth performance						
Initial body weight (kg)	17.8	17.6	17.6	17.8	0.1	0.357
Final body weight (kg)	21.5 ^a^	20.8 ^b^	20.7 ^b^	20.9 ^b^	0.14	0.020
Weight gain (g/day)	40.7 ^a^	37.1 ^ab^	34.4 ^b^	34.2 ^b^	1.4	0.025
Feed-to-gain ratio	16.6	17.9	18.2	18.6	2.33	0.506
Carcass characteristics						
Slaughter body weight (kg)	21.5 ^a^	20.8 ^b^	20.7 ^b^	20.9 ^b^	0.15	0.020
Hot carcass weight (kg)	7.7	7.1	6.9	7.2	0.17	0.056
Empty body weight (kg)	16 ^a^	15.4 ^b^	15.3 ^b^	15.4 ^b^	0.16	0.045
Dressing percentage (%)	35.7	34.2	33.5	34.4	0.6	0.116
Rib eye area (cm^2^)	8.0	7.5	7.3	7.3	0.3	0.256
Edible offal						
Blood (g)	1004	1034	1059	973	34	0.343
Liver (g)	305	290	302	273	10	0.175
Kidney(g)	71	70	74	69	2	0.547
Heart (g)	102	109	102	95	5	0.297
Tongue (g)	73	66	71	69	3	0.428
Reticulo-rumen (g)	599	614	589	697	43	0.315
Abomasum-omasum (g)	353	358	351	349	5	0.681
Small intestine (g)	470	437	423	432	29	0.687
Large intestine (g)	507	518	502	506	19	0.940

^a,b^ Different superscripts indicate significant differences at *p* < 0.05.

## Data Availability

The datasets generated during and/or during the current study are available from the corresponding author on reasonable request.

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
