# Peer review of "Effect of Barley Variety on Feed Intake, Digestibility, Body Weight Gain and Carcass Characteristics in Fattening Lambs"

_animals, 2021, doi:10.3390/ani11061773_

Round 1
Reviewer 1 Report
Manuscript ID: animals-1171378
General
The purpose of this study was to evaluate four varieties of barley straw included (~50%) in growing diets for lambs on total tract digestibility, growth performance and carcass traits. Even though this reviewer acknowledges the hard work devoted by the authors to the present experience, I also consider that the manuscript has a deficiency that limit it consideration for publication in Animals journal, at least in its current presentation
A concern is the number of observation within treatments. Even when the number of repetitions by treatment is enough (n=5), the number of observations within treatment is lower than the minimum recommended (n=8) for growth performance trials considering the intrinsic CV for ADG.
Materials and methods are not enough detail described. More proper description of treatments is necessary.
Results: Some data must be checked and statistically significance (p-values) must be revised
Discussion: Some aspect of discussion are imprecise and others must be aligned when results be checked and, is in case, corrected.
Specific
Title: Ok
Simple summary: Must be improved.
L11-15: Please, rewording it sometime like: Using a native barley straw as reference, the barley straw from some improved varieties were tested on digestibility and performance in lambs. Significant differences were observed on feed intake, digestibility, body weight gain and feed conversion among lambs fed straws from different barley varieties, pointing to the importance of genetic variation in the feeding value of barley straw.
Through the document you describe the experimental units as "sheep" "rams" "lambs". Please, use a specific term to describe experimental units (due the initial weight of lambs preferable use "lambs")
Abstract: Incomplete description, must improve.
L16-21: Rewording as follow: Twenty lambs (18 ± 0.22 kg initial weight) were blocking by weight and individually assigned into pens in order to evaluate the effects of barley straw variety on digestibility, performance and carcass characteristics. Four treatments were tested: 1) local straw barley (as control), 2) HB1963 (high grain and straw yields), 3) Traveller (high straw yielder), and 4) IBONE 174/03 (high grain yielder). A concentrate (50:50 wheat bran and noug seed cake) was offered constant (300g), while the straw was offered ad libitum. Digestibility trial lasted 22-d (15-d to adapt dietary treatments and 7-d for sampling). The growth-performance trial lasted 90-d, at the end of trial all lambs were harvested and carcass were evaluated.
L22-28: The results should be checked, once you check them and if necessary, they are corrected, then incorporate it into the abstract
Materials and methods.
Mat & Methods are not enough detail described. More proper description of treatments is necessary.
L85-86: Specify vaccines and dewormed use!
L88: Describe characteristics of individual pens!
L90-92: The aim of this experiment was compared straws from improved varieties of barley. For that, you need a Control straw (in this case the straw from local genotype barley). Thus, description of treatments must: 1) control, and 2), 3), and 4) the test straws. It's important to note that concentrate was offered at constant quantity and straw was offered ad libitum (this specification is important)
Could be? Four treatments were tested: 1) local straw barley (as control), 2) HB1963 (high grain and straw yields), 3) Traveller (high straw yielder), and 4) IBONE 174/03 (high grain yielder). A concentrate (50:50 wheat bran and noug seed cake) was offered constant (300g), while the straw was offered ad libitum
L100: Specify here: During sample period lambs were equipped with collection bags (indicate the type and trade mark of bags, if correspond).
L110: Start the sentence as: The growth-performance trial was conducted 10 days after the digestion experiment was concluded.
Questions that need to be answered and specified in text:
1) Each lambs were fed in growth trial with the same treatment that received during digestion trial? (i.e., lamb #4 received treatment 2 during digestion trial and in growth-performance trial received the same treatment), or lambs were weighed, blocking and newly randomly assigned to treatments?
2) For both, specify model, city and country of the scale used to weight animals and the balance used to weight feed and refusals.
L115: Please include calculations for ADG and for feed-to-gain ratio (use FG calculi instead feed conversion).
L121: remove “experimental rams”, write as: all lambs were..
L122: “Lambs were individually weighed before slaughter” “Carcass variables were registered individually”.
L123-127: Very poor description!!. Please include time and temperature of cooling before take the variables and what procedure (and its reference) was utilized to take measures of each carcass variable
L132: Specify reference for NDF determination, and if exist some modification of determination describe it.
Results
L141: Did you mean experimental diet?
L142-144: Do not include logical results like expose in this paragraph (remove). Instead this, expose results of chemical differences of local straw vs tested varieties.
L146-152: Results was written carelessly and must be improved. The wording is somewhat confusing, it requires more ordering and a more adequate focus on aim of the study.
You should consider the following to interpret better your results. Barley straw was offered ad libitum while intake of concentrate remain constant. If were differences on barley straw intake between treatments, thus, the concentrate: barley straw ratio was different. Then, the differences on digestibility coefficient are due by differences on proportion of concentrate:straw intake rather than effects of straws per se. A clear example is the protein digestibility of "local straw". You argue that (L150) "whereas crude protein digestibility was higher with the local variety compared to the other varieties". This is not totally true when you compare to IBON174-03 and TRAVELER varieties. In those cases, the greater CP digestibility observed for this treatment was a reflection of a lower straw intake (325 g/d) and lower protein content (4.3%) in local straw. I explain about it, in local-straw treatment lambs consumed 14 g of protein from straw and 61g from concentrate (19% of low-digestible protein of total protein consumed), while lambs fed, for example, treatment IBON174-03 consumed 20.5 g of protein from straw and 61 g CP from concentrate (25% of low-digestible protein of total protein consumed) As you can noted, the lambs of treatments IBON174-03 and TRAVELLER consumed more indigestible protein during the experiment, which explains the result. And not was because protein of straws from varieties were less digestible than local straw. Even so, I don’t think that protein digestibility of local are statistically different from IBONI and TRAVELLER. (SEM=1.94, numerically differences<2.2) Please check this.
Growth performance
Include in results intake data observed during growth-performance trial. Use gain-to-feed ratio instead “fed conversion”
L16: Discussion. Must be improved
Include in discussion the chemical composition of tested straws! (i.e. why are not similar, and compare with others reports).
L165-169: The values discussed here not represent the digestibility of barley straw, represent the digestibility of complete diet (concentrate plus straw). Furthermore, since straw intake was not equal between treatments (because straw intake was planned as libitum) this comparison are inaccurate. This type of misjudgment is common, but imprecise to adequately discuss one of the central objectives of the study (compare digestibility between barley straws tested)
I allow myself to explain a procedure to estimate more accurately digestibility of straws. According to DM digestibility (DMd) for wheat bran (76%) and noug seed cake (86%) reported previously (Haddad et al., 2000; Paya et al., 2007, Ribero et al., 2017) then, DMd for concentrate used in this trial is 80.5%. Based on this value, and data of DM intake for concentrate and straw, and DM total tract digestibility, estimate DM digestibility of straw of each treatment can be estimated as follows: ((DM intake* digestibility DM) -(0.805*300))/straw intake. Straw intake was 325, 312, 356, and 372 g for Local, HB1963, TRAVELLER, and IBON174, respectively. In such way that digestibility of local straw resulted in 48%, this value is in close agreement for digestibility of barley straw determined in previous report (Hassan et al., 2012). The estimated DM digestibility for HB1963, TRAVELLER, and IBON174 are 56.5, 52.3, and 64.9%, respectively. If you agree, use this values to generate an appropriate discussion.
L181-188: Are you sure that lambs fed "local straw" reach the maximal capacity of intake? Please estimate the maximal capacity for each group according to chemical composition of diets tested and include this in discussion [you can check Pinto de Oliveira et al. (2020) PLOsONE15(12): e0244 or Cannas et al 2004]
L199: Use fed-to-gain ratio instead fed conversion
L210: Take care whit this generalization. In this experiment it was observed a low dressing percentage because the slaughter weight (very light) and by energy density of experimental diet (very low, <1.7 Mcal NEm). In fattening lambs slaughter at heavy weight (i.e. 55-60 kg) feeding with high-energy diets (~2.20 Mcal/NEm) during long-term fattening (112-d) you can registered 57-59% dressing percentage. Thus, not generalize.
L219: How much low? Be specific
L226: This study was planned to compare 4 sources of barley straw, the effects registered were digestibility, some variables of growth and performance and some carcass characteristics, the experimental units were lambs. How this study show the differences on yield productivity of barleys?? Please, focus in your objectives.
Conclusions
Rewording all conclusions. Please, focus in your findings according to the object of the study.
Table 3.
Please check values of OM shown in Table 3. According to the ash concentration of each ingredient (Table2) and daily intakes of concentrate (300 g) and straw (372, 356, 312 and 325 g/d for IBON174-03, TRAVELLER, HB1963, and Local straw (Table3), then, the ash intakes (g/d) were 79.3, 75.1, 72.1 and 71.6. This result on daily OM intake of 593, 581, 540, and 553 g. those values are not in agreement to those shown in Table.
I don’t think that protein digestibility of local are statistically different from IBONI and TRAVELLER. (SEM=1.94, numerically differences<2.2) Please check this.
Table 4.
Include intake data observed during growth-performance trial. Use gain-to-feed ratio instead “fed conversion”
Author Response
Dear reviewer
Thank you for your valuable and details comment to improve our MP. We have carefully reviewed the comments and have revised the MP accordingly. our responses are given in a point by point manner in the 2nd column of the rebuttal letter shown in yellow background.
With best regards
Mulugeta Tilahun Keno

Reviewer 2 Report
The english language must be revised by an english mother tongue scientist, there are many mistakes and/or not appropriate terms.
Title
line 3: write "fattening rams" not sheep.
Use always rams allover the manuscript, not sheep.
Abstract
line 21: delete one dot after "112 days".
line 22: specify in brackets the length of the trial before "...for carcass characteristics..."
line 22: insert the level of statistical difference (P<0.05) after ".. significantly higher".
line 28: delete one dot after ".. barley variety".
Introduction
line 33: the sentence "... but is more recently grown..." is completely unclear, rewrite it carefully.
line 37: why Rejected has the capital R???
Material and methods
line 124: including included????
Author Response
Dear reviewer
Thank you very much for your comments that helped us to improve our MP
We reviewed the MP accordingly and our responses are given in a point by point manner in in the 2nd column of the rebuttal letter in yellow background .
With best regards
Mulugeta Tilahun Keno

Round 2
Reviewer 1 Report
Thanks to the authors for their efforts to improve the quality of the paper. The authors basically met my review requirements. Only one change. Please, in "Statistical analysis" describe as: The experimental lambs were blocked according.." instead "The experimental rams were blocked according.."
Reviewer 2 Report
The Authors have strictly followed the suggestions received by the Reviewer, the manuscript can now be accepted for publication.